# Attitude and predictors of exclusive breastfeeding practice among mothers attending under-five welfare clinics in a rural community in Southwestern Nigeria

Tope Michael Ipinnimo[1]*, Olanrewaju Kassim Olasehinde[1], Taofeek Adedayo Sanni[1,2], Ayodeji Andrew Omotoso[1], Rita Omobosola Alabi[1], Paul Oladapo Ajayi[3], Kayode Rasaq Adewoye[1,2], John Olujide Ojo[1,2], Olayinka Oloruntoba[4], Ademuyiwa Adetona[2], Mojoyinola Oyindamola Adeosun[3], Temitope Moronkeji Olanrewaju[5], Oluseyi Adedeji Aderinwale[6], Blessing Omobolanle Osho[7], Adewumi Rufus Fajugbagbe[7], Precious Aderinsola Adeyeye[7], Ayotomiwa Fiyinfoluwa Ajayi[7]

1 Department of Community Medicine, Federal Teaching Hospital, Ido-Ekiti, Nigeria, 2 Department of Community Medicine, Afe Babalola University, Ado-Ekiti, Nigeria, 3 Department of Community Medicine, Ekiti State University, Ado-Ekiti, Nigeria, 4 Department of Surgery, Federal Teaching Hospital, Ido-Ekiti, Nigeria, 5 Department of Family Medicine, Federal Teaching Hospital, Ido-Ekiti, Nigeria, 6 Department of Obstetrics and Gynaecology, Federal Medical Centre, Abeokuta, Nigeria, 7 Department of Public Health, Kwara State University, Malete, Nigeria

* abbeymagnus@yahoo.com

**Data Availability Statement:** All relevant data are within the manuscript and its Supporting Information files.

## Abstract

### Background

Much previous research on exclusive breastfeeding has focused on urban and semi-urban communities, while there is still a paucity of data from rural areas. We assessed the attitude and practice of exclusive breastfeeding and its predictors among mothers attending the under-five welfare clinics in a rural community.

### Methods

A cross-sectional study was conducted among consecutively recruited 217 mothers attending the three health facilities under-five welfare clinics in Ido-Ekiti, Southwest, Nigeria. Information was collected with a semi-structured interviewer-administered questionnaire adapted from previously published research works. Descriptive and inferential statistics were carried out using IBM SPSS Statistics for Windows, Version 26.0.

### Results

More than half of the mothers, 117(53.9%) were ≥30 years old, and 191(88.0%) were married. Almost all, 216 (99.5%) attended an ante-natal clinic; however, 174(80.2%) delivered in the health facility. The respondent's mean ± SD exclusive breastfeeding attitudinal score was 29.94 ± 2.14 (maximum obtainable score was 36), and the proportion of mothers that practiced exclusive breastfeeding was 40.6%. Married mothers were more likely to practice exclusive breastfeeding than their unmarried counterparts (AOR:6.324,

**Funding:** The author(s) received no specific funding for this work.

**Competing interests:** The authors have declared that no competing interests exist.

95%CI:1.809–22.114). The common reasons for not practicing exclusive breastfeeding were work schedule 57(26.3%), cultural beliefs and the need to introduce herbal medicine 32(14.7%), and insufficient breast milk 30(13.8%).

## Conclusion

This study revealed a good disposition with a suboptimal practice towards exclusive breastfeeding. Also, being married was a positive predictor of exclusive breastfeeding. Therefore, we recommend policies that will improve exclusive breastfeeding among mothers in rural areas, especially those targeting the unmarried, to achieve the World Health Organization's target.

## Introduction

Breastmilk is described as the primary source of nutrition for newborns and provides all the nutritional needs for the first few months of life [1]. It is safe, clean and essentially contains water, nutrients, and antibodies in adequate proportions to promote child growth and development [2]. Exclusive breastfeeding (EBF) involves feeding only breastmilk to a child for the first six months of life, except for medically prescribed drugs or supplements [3]. The World Health Organisation (WHO) recommends EBF because it is cost-effective, and significantly lower the risk of diarrhoea, malnutrition as well as morbidity and mortality among the under-five age group [2]. Breastfeeding not only proves beneficial to child health but also contributes to maternal health as well as providing social and economic benefits [4].

Globally, the average EBF practice between 2015–2021 was 48% [5] and in Italy, only 33.3% practiced EBF. More than half of the Italian women had heard about EBF, with the majority of them believing that EBF is important to both mother and child [6]. However, a study in Bangladesh, which focused on mothers in rural areas, showed poor knowledge (34.5%) and practice (27.9%) of EBF [7]. In East Africa, a systematic review revealed that 42% of mothers preferred to feed their infants with breastmilk alone for the first six months of life, while 55.9% of them had practiced EBF for at least six months [8]. The West Africa region had a prevalence of 35.0% for EBF between 2015–2021, one of the lowest in the world for the same period [5].

In Nigeria, only 28.7% of babies were exclusively breastfed in 2018 [9]. This was even lower in the rural population, with a report of 24.3% [9]. A study in a semi-urban area of Sokoto state, Northwest Nigeria, indicated that only 31.0% of women practiced EBF [10]. Another study conducted in one of the urban centers in Abuja, the Federal Capital Territory of Nigeria showed that 54.4% had practiced EBF despite a significant positive attitude determined by the study (70% agreed that EBF was adequate for their children) [11]. A similar study in another urban center in Benin, Edo state, South-south Nigeria, showed an "ever-breastfed rate" of 100.0%, but only 40.7% practiced EBF [12]. A survey carried out in two tertiary hospitals in Ogun state, Southwest Nigeria, revealed that more than half (58.8%) of the women practiced EBF [13].

Optimal breastfeeding practices are important in preventing and managing malnutrition in children [2]. Stunting, which is a form of malnutrition and an indication of poor nutrition beginning in utero into early childhood, affecting an estimated 144 million (21.3%) under-five children globally in 2019 with the highest burdens in Africa and Asia continents [14]. Over one-third (37.0%) of Nigerian children aged 6–59 months are stunted, and the prevalence of stunting is almost twice as high among children in rural areas (45.0%) as among those in urban areas (27.0%) [9].

Many previous studies on EBF have centered on urban and semi-urban communities, while there is still a paucity of data from rural areas [10–13]. To the best of the authors' knowledge, there has not been a study of this nature assessing the attitude and practice of mothers on EBF in our environment. This research was designed to identify the gap in the attitude and practice of mothers living in this area toward EBF as it may aid strategy development in improving the nutritional status of infants and children living in the communities.

This study aims to determine the attitude, and practice as well as identify the predictors of practice of EBF among mothers attending the under-five welfare clinics in a rural community in Southwest Nigeria. The findings of this work would contribute to the literature and can help facilitate further research on the subject. Also, it would aid in defining the gaps in women's attitude to EBF and its practice, especially in the rural areas. This could then serve as a basis for health planning, policy formulation and implementation to improve EBF practice in rural Nigeria in order to meet the World Health Organization (WHO) recommended target of at least 50.0% EBF rate in all communities by 2025 [15].

## Methods

### Study area and design

This was a cross-sectional study carried out between 1st and 30th November 2021 in all the under-five welfare clinics in Ido-Ekiti, Southwest, Nigeria. Ido-Ekiti is a town with an estimated population of 37,000 [16]. There are three health facilities with under-five welfare clinic in the community which included the Basic Health Centre (BHC), Comprehensive Health Centre (CHC) and the Federal Teaching Hospital (FTH), Ido-Ekiti. The BHC and CHC are primary health facilities, each with a staff strength of less than thirty healthcare workers mainly registered nurses/midwives and community health extension workers. The BHC and CHC under-five welfare clinics provides services such as immunization and growth monitoring on clinic days which is just once in a week and each of them sees an average of 10 to 20 children per clinic day. The FTH is a tertiary health facility that serves as a referral centre for other health facilities within the environs. The under-five welfare clinic in FTH is located at the Department of Community Medicine of the hospital and it offers services every day with the exception of weekends (Saturday to Sunday) to an average of 20 children per day. The staff in the clinic consists of community/public health physicians, public health nurses and community health extension workers.

### Participants, sample size determination and sampling technique

The study population consisted of mothers attending the under-five welfare clinics in the community. The study included all healthy mothers with healthy baby(ies) attending the clinics. Mothers with adopted or fostered babies were excluded from the study. The minimum sample size for the study was determined using Leslie Fisher's formula [17]. A 95% confidence interval, and 5% degree of accuracy was assumed. A sample size of 217 was obtained after using the prevalence of EBF from a previous study [18]. All the three health facilities were used for the study. The number of mothers selected from the health facilities was determined using proportionate allocation based on the under-five welfare clinic attendance. Eligible mothers attending the clinic were recruited consecutively into the study until the sample size was achieved.

### Data collection methods

Researchers administered the questionnaire in a scheduled area of the health facilities through face-to-face interviews with the mothers immediately after their clinics.

## Data collection tools

The study instrument used to collect information from the women is a 26-item semi-structured interviewer-administered questionnaire adapted from previously published research works [19, 20]. The questionnaire obtained data on socio-demographic and other (such as antenatal clinic attendance, place of delivery) variables, attitude towards EBF, practice of EBF and reasons for not breastfeeding exclusively. The questionnaire was pre-tested on a sample (twenty) of mothers in a CHC in another community which were neither analyzed nor included in the study. Necessary adjustments and corrections were made to the questionnaire after the pre-test. The items in the study instrument were tested for internal consistency using the Cronbach coefficient alpha test and a score of 0.9 was obtained. Twelve questions assessed the attitude of mothers on EBF. Mothers were asked to rate their responses on a 3-point Likert scale (1 to 3) measuring the intensity of mother's attitudes [19]. A positive response was rated 3, a neutral was rated 2, and a negative was rated 1. The attitudinal score of each respondent was estimated by summing their rating with a maximum obtainable score of 36. Exclusive breastfeeding practice was assessed with a question inquiring if mothers have given only breast milk to the baby for the first 6 months without infant formula or any other product unless it was prescribed by a physician.

## Variables

The independent variables in this study were sociodemographic variables, maternal characteristics such as antenatal clinic attendance, place of delivery and mode of delivery, while the dependent variable included the attitude and practice of EBF among the mothers.

## Data management and statistical analysis

The data collected were entered, cleaned and analysed using computer software IBM SPSS (IBM SPSS Statistics for Windows, Version 26.0. Armonk, NY: IBM Corp). Data were summarized in tables using frequencies and percentages. The attitudinal scores of the participants were calculated and presented using mean and standard deviation. The mean attitudinal scores across each socio-demographic and maternal variables category was compared using the independent Student-T-test. The association between the mother's characteristics and the practice of EBF was assessed using the Chi-square test. Binary logistic regression analysis was used to determine the predictors of EBF. The logistic regression model was built with variables with $p \leq 0.2$ at the bivariate (Chi-square test) level of analysis. P-values <0.05 were considered significant for the inferential statistics.

## Ethical considerations

Ethical approval (ERC/2021/10/18/937A) for this study was obtained from the Human Research and Ethics Review Committee of the Federal Teaching Hospital, Ido-Ekiti, Nigeria. Permission was obtained from the officer-in-charge of the BHC and CHC as well as the consultant-in-charge of the FTH clinics. Additionally, written informed consent to participate was obtained from the mothers before the interviews. Confidentiality and anonymity were maintained by not collecting personal data such as names and phone numbers from the respondents.

## Results

Table 1 shows that the mean age (±standard deviation) of the mothers was 30.5 (±6.4) years. More than half of the mothers, 117 (53.9%) were ≥ 30 years and more than two-thirds, 148

**Table 1. Socio-demographic and other characteristics of the respondents (N = 217).**

| Variable | Frequency | Percent |
|---|---|---|
| | (N = 217) | (%) |
| **Age** | | |
| < 30 years | 100 | 46.1 |
| ≥ 30 years | 117 | 53.9 |
| *Mean age (±Standard deviation)* | *30.5 (±6.4)* | |
| **Level of education** | | |
| Secondary | 69 | 31.8 |
| Post-secondary | 148 | 68.2 |
| **Marital status** | | |
| Unmarried | 26 | 12.0 |
| Married | 191 | 88.0 |
| **Religion** | | |
| Christianity | 174 | 80.2 |
| Islam | 43 | 19.8 |
| **Tribe** | | |
| Yoruba | 179 | 82.5 |
| Hausa | 16 | 7.4 |
| Igbo | 22 | 10.1 |
| **Occupation** | | |
| Farmer/trader/artisan | 76 | 35.0 |
| Civil servant | 94 | 43.3 |
| Professional | 20 | 9.2 |
| Student | 27 | 12.5 |
| **ANC attendance during the pregnancy of this baby** | | |
| Yes | 216 | 99.5 |
| No | 1 | 0.5 |
| **If yes, were you informed on exclusive breastfeeding (n = 216)** | | |
| Yes | 214 | 99.1 |
| No | 2 | 0.9 |
| **Place of delivery of this baby** | | |
| Outside health facility | 43 | 19.8 |
| Within health facility | 174 | 80.2 |
| **Mode of delivery of this baby** | | |
| Vaginal delivery | 182 | 83.9 |
| Caesarean section | 35 | 16.1 |
| **Birth order of the baby** | | |
| First | 91 | 41.9 |
| Second | 75 | 34.6 |
| 3 and above | 51 | 23.5 |

(68.2%), held a post-secondary school degree. More than three-quarters, 191 (88.0%) and 179 (82.5%) were married and of Yoruba ethnicity respectively. Ninety-four (43.3%) of them were civil servants. The majority 216 (99.5%) of mothers attended the ante-natal clinic, out of which 214 (99.1%) were informed on EBF. Most, 174 (80.2%) delivered within a health facility and 182 (83.9%) deliveries were via vaginal delivery. Ninety-one (41.9%) mothers are currently nursing their first child.

Table 2 shows that the mean (standard deviation) EBF attitudinal score for all respondents was 29.94 (2.14) out of 36 maximum possible scores. EBF attitudinal scores significantly differed across age (T = -2.844, p = 0.005), level of education (T = -2.273, p = 0.025), ante-natal clinic attendance during the pregnancy of the baby (T = 4.895, p < 0.001), been informed on

**Table 2. Exclusive breastfeeding attitudinal score associated with respondents' characteristics (N = 217).**

| Variable | Attitudinal score | Test | p-value |
|---|---|---|---|
| | Mean ± SD | | |
| **All respondents** | 29.94 ± 2.14 | - | - |
| **Age (in years)** | | -2.844[T] | **0.005** |
| < 30 years | 29.50 ± 1.956 | | |
| ≥ 30 years | 30.30 ± 2.226 | | |
| **Level of education** | | -2.273[T] | **0.025** |
| Secondary | 29.43 ± 2.29 | | |
| Post-secondary | 30.16 ±2.02 | | |
| **Marital status** | | -0.480[T] | 0.634 |
| Unmarried | 29.76 ± 1.839 | | |
| Married | 29.95 ± 2.180 | | |
| **Religion** | | 1.974[T] | 0.053 |
| Christianity | 30.09 ± 2.007 | | |
| Islam | 29.27 ± 2.529 | | |
| **Tribe** | | 0.317[F] | 0.729 |
| Yoruba | 29.97 ± 2.075 | | |
| Hausa | 30.00 ± 1.591 | | |
| Igbo | 29.59 ± 2.938 | | |
| **Occupation** | | 0.427[F] | 0.734 |
| Farmer/trader/artisan | 29.72 ± 2.194 | | |
| C/S | 30.09 ± 2.374 | | |
| Professional | 30.00 ± 2.000 | | |
| Student | 29.92 ± 0.196 | | |
| **ANC attendance during the pregnancy of this baby** | | 4.895[T] | **<0.001** |
| Yes | 29.98 ± 2.034 | | |
| No | 20.00 ± 0.000 | | |
| **If yes, were you informed on exclusive breastfeeding** | | 14.378[T] | **<0.001** |
| Yes | 30.00 ± 2.034 | | |
| No | 28.00 ± 0.000 | | |
| **Place of delivery** | | 0.571[T] | 0.570 |
| Outside health facility | 30.11 ± 2.372 | | |
| Within health facility | 29.98 ± 2.083 | | |
| **Mode of delivery** | | -2.773[T] | **0.008** |
| Vaginal delivery | 29.78 ± 2.166 | | |
| Caesarean section | 30.74 ± 1.820 | | |
| **Birth order** | | 9.533[F] | **0.001** |
| First | 29.45 ± 2.088 | | |
| Second | 30.77 ± 1.713 | | |
| Third and above | 29.56 ± 2.443 | | |

SD: Standard deviation

[T]: Student T test

[F]: Analysis of variance

EBF during ante-natal clinic (T = 14.378, p < 0.001), mode of delivery (T = -2.773, p = 0.008) and the birth order of the child (F = 9.533, p = 0.001). The attitudinal score was better among mothers aged ≥ 30 years, those with post-secondary education, mothers with antenatal clinic attendance, those informed on EBF, mothers who had caesarian section delivery, and those with higher birth orders. The attitudinal score did not significantly differ across other variables.

Eighty-eight (40.6%) mothers practiced EBF (Table 3). According to Fig 1, the reasons for the mothers not practicing EBF were work schedule 57 (26.3%), cultural beliefs and the need to introduce herbal medicine 32 (14.7%), insufficient breast milk 30 (13.8%), baby getting hungry and thirsty 28 (12.9%), HIV and unplanned pregnancy 19 (8.8%), early and single motherhood 14 (6.5%) and other reasons 4 (1.8%) [Fig 1]. Table 3 shows that there was a statistically significant association between marital status and the practice of EBF (X = 8.993, p = 0.003). A higher proportion of married mothers practiced EBF than unmarried mothers (married: 44.5%; unmarried: 11.5%). There was no statistically significant association between other respondents' variables and the practice of EBF. Binary logistic regression in Table 4 revealed that married mothers were about 6 times more likely to practice EBF than those who were unmarried (AOR:6.324, 95%CI: 1.809–22.114).

## Discussion

This study assessed the attitude towards EBF, its practice, and predictors among mothers attending under-five welfare clinics in a rural community. The mean attitude score towards EBF for all mothers in this study was 29.9 out of 36 maximum possible scores. This is 83.2% of the maximum obtainable score which depicts a good disposition towards EBF among the mothers. This finding is higher than what was obtained in rural communities of Lagos, Nigeria [20]. However, it is similar to what was found in studies done among lactating mothers in a rural Ghana community and Ethiopia [19, 21]. This level of attitude is good for the mothers, although there is room for improvement.

This study revealed that EBF attitude was positively associated with age, as mothers who were 30 years and over had a better attitude than those aged less than 30 years. This finding is consistent with findings from studies in Ondo state, Southwest, Nigeria [22] and in the Island of Abu Dhabi, United Arab Emirates [23]. Also, mothers who had attended the antenatal clinic during the pregnancy of the baby had a better attitude than mothers who did not attend the clinic. The antenatal clinic provides nutritional counseling and education including the health benefits of EBF and other infant feeding practices as part of the health promotion measures during visits. Unsurprisingly, this exposure not only improves maternal attitude but subsequently brings about the likelihood of better practice as reported in a study done in Rawalpindi [24].

This study found that mothers with a higher level of education had better attitude toward EBF. Maternal level of education is a significant determinant of infant feeding practices, and this is supported by a study done in Ondo state, Southwest, Nigeria which found that attitude towards EBF is higher among lactating mothers with higher levels of education [22]. Mothers with higher levels of education can understand and quickly realize the benefits of EBF to their infants and are more motivated to practice it. Programs promoting EBF should be adapted to appeal to mothers who have lower levels of education. The maternal attitude of EBF was better among mothers with 2nd birth than those with their first birth. This is similar to the findings in a study done in Ibadan, Southwest, Nigeria [25]. This finding is not surprising as multiparous mothers have previously developed a prior experience and skillset allowing for better EBF attitude in subsequent deliveries.

**Table 3. Association between respondents' characteristics and practice of exclusive breastfeeding (N = 217).**

| Variable | Exclusive breastfeeding | | | |
|---|---|---|---|---|
| | Yes | No | | |
| | (n = 88) (%) | (n = 129) (%) | Test | p-value |
| **All respondents** | 88 (40.6) | 129 (59.4) | - | - |
| **Age group (years)** | | | $0.015^X$ | 1.000 |
| <30 years | 41 (41.0) | 59 (59.0) | | |
| >30 years | 47 (40.2) | 70 (59.8) | | |
| **Level of education** | | | $0.091^X$ | 0.769 |
| Secondary | 29 (42.0) | 40 (58.0) | | |
| Post-secondary | 59 (39.9) | 89 (60.1) | | |
| **Marital status** | | | $8.993^Y$ | **0.003** |
| Unmarried | 3 (11.5) | 23 (88.5) | | |
| Married | 85 (44.5) | 106 (55.5) | | |
| **Religion** | | | $0.249^X$ | 0.729 |
| Christianity | 72 (41.4) | 102 (58.6) | | |
| Islam | 16 (37.2) | 27 (52.8) | | |
| **Tribe** | | | $3.506^F$ | 0.173 |
| Yoruba | 75 (49.1) | 104 (58.9) | | |
| Hausa | 3 (18.8) | 13 (81.3) | | |
| Igbo | 10 (45.5) | 12 (54.5) | | |
| **Occupation** | | | $2.257^X$ | 0.529 |
| Farmer/trader/artisan | 32 (42.9) | 44 (57.1) | | |
| C/S | 34 (36.2) | 60 (63.8) | | |
| Professional | 8 (40.0) | 12 (60.0) | | |
| Student | 14 (51.9) | 13 (48.1) | | |
| **ANC attendance during the pregnancy of this baby** | | | $<0.001^Y$ | 1.000 |
| Yes | 88 (40.7) | 128 (59.3) | | |
| No | 0 (0.0) | 1 (100.0) | | |
| **If yes to question 7, were you informed on exclusive breastfeeding** | | | $0.207^Y$ | 0.649 |
| Yes | 88 (41.1) | 126 (58.9) | | |
| No | 0 (0.0) | 2 (100.0) | | |
| **Place of delivery of this baby** | | | $0.715^X$ | 0.488 |
| Outside HF | 15 (34.9) | 28 (65.1) | | |
| Within HF | 73 (42.0) | 101 (58.0) | | |
| **Mode of delivery** | | | $0.005^X$ | 1.000 |
| Vaginal delivery | 74 (40.7) | 108 (59.3) | | |
| Caesarean section | 14 (40.0) | 21 (60.0) | | |
| **Birth order** | | | $2.859^X$ | 0.245 |
| First | 32 (35.2) | 59 (54.8) | | |
| Second | 36 (48.0) | 39 (52.0) | | |
| Third and above | 20 (39.2) | 31 (60.8) | | |
| **Mean attitudinal score** | 30.09 ± 1.891 | 29.82 ± 2.295 | $0.883^T$ | 0.378 |

$^X$: Pearson Chi-Square

$^Y$: Continuity correction

$^F$: Fisher's Exact Test

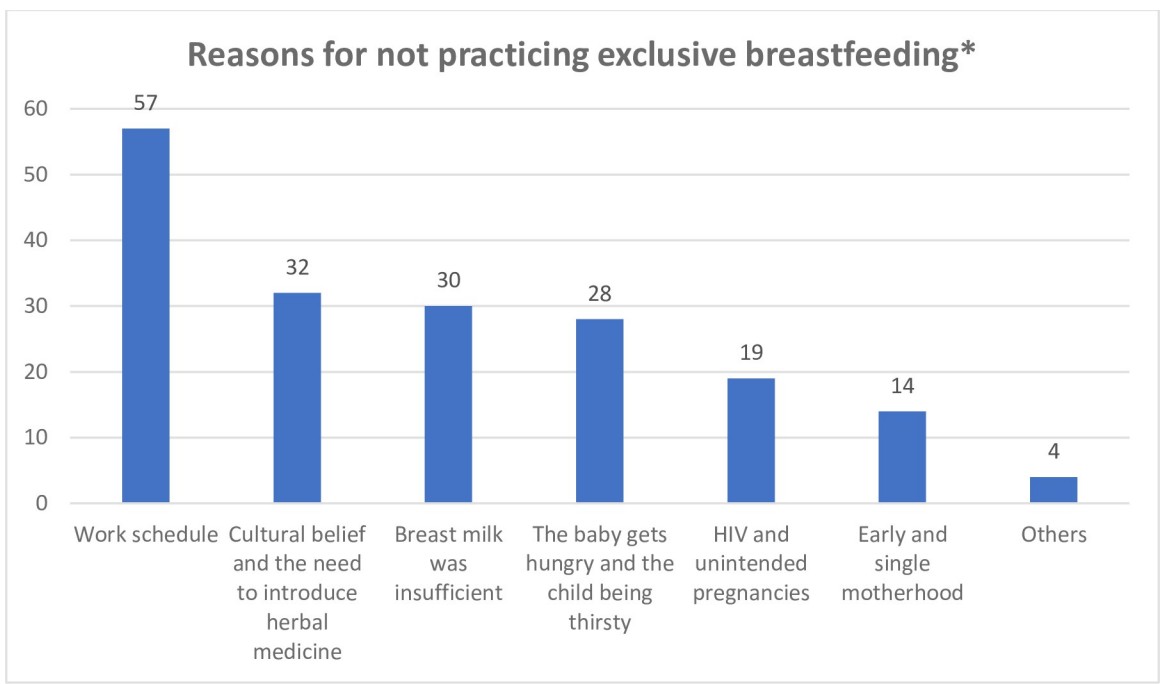

**Fig 1. Reasons for not practicing exclusive breastfeeding (N = 217).** *Multiple response.

Less than half (40.6%) of mothers in this study practiced EBF. This is similar to a study done in Edo state, South-south, Nigeria with a prevalence of 40.7% [12]. However, it is higher than the findings in previous studies done in Italy (33.3%), Bangladesh (34.5%), Southern Ethiopia (14.9%), Jigawa (26.8%) and Sokoto (31.0%) states, Northern, Nigeria (31.0%) [6, 7, 10, 26, 27] and lower than the prevalence recorded from studies conducted in Debre Berhan District, Ethiopia (68.6%), urban centers of Abuja (54.4%), Ikorodu, Lagos (69.4%) and in Ogun State, Southwest, Nigeria (58.8%) [11, 13, 20, 28]. Interestingly, the prevalence of practice of EBF in this study aligns with the WHO targets of at least 50.0% rate in all communities whether rural or urban by 2025 [15]. Though, close to the WHO target, several barriers still hinder the practice of EBF. Work schedule, cultural beliefs and the need to introduce herbal medicine, insufficient breast milk, baby gets hungry and thirsty, HIV and unplanned pregnancy, early and single motherhood were mentioned by the mothers as barriers to EBF. Some of these barriers have equally been reported in previous studies [29, 30].

**Table 4. Predictors of practice of exusive breastfeeding.**

| Predictors | B | P-value | AOR | 95% CI for AOR | |
|---|---|---|---|---|---|
| | | | | Lower | Upper |
| **Marital Status** (Unmarried)[Ref] | | | 1.000 | | |
| **Marital Status** (Married) | 1.844 | **0.004** | 6.324 | 1.809 | 22.114 |
| **Tribe** (Yoruba)[Ref] | | | 1.000 | | |
| **Tribe** (Hausa) | 0.357 | 0.459 | 1.429 | 0.556 | 3.674 |
| **Tribe** (Igbo) | 1.429 | 0.072 | 4.173 | 0.879 | 19.803 |

B: Intercept, AOR: Adjusted odd ratio, CI: Confidence interval

It was observed from this study that mothers who are married were about 6 times more likely to practice EBF than those who are unmarried. Expectedly, this result is substantiated by another study in Ethiopia [31]. This may be due to the responsibility-sharing and support provided by the husband that could aid the practice of EBF. However, this finding contrast that found in a study done among first-time mothers in Ethiopia which revealed that participants who were not married were almost 3-fold more likely to practice EBF compared with married counterpart [32]. This difference could be due to birth-related traditional practices of some Ethiopian where first-time mothers at 8 months gestational age visit their parent's home in preparation for births, and afterward, stay for 40 days after birth with their infant and never left alone. Against these background, one can deduce that a good support system from husbands of married women or from parents of unmarried women post-delivery makes the practice of EBF more feasible.

## Strengths and limitations

To the best of our knowledge, this is the first study in Ekiti state that assessed the attitude and practice of exclusive breastfeeding among rural mothers. We also identified predictors of exclusive breastfeeding among this population.

However, the study shares the limitations of the cross-sectional study design. In addition, the research is prone to social desirability and recall bias since questions were asked about the past experiences of the mothers on breastfeeding. Lastly, this health facility-based study that was carried out in the under-five welfare clinic and might have left out mothers who do not seek healthcare in health facilities or patronize other institutions such as patent medicine stores, drug hawkers and herbalists for their healthcare.

## Conclusion and recommendation

The study revealed a good disposition towards EBF among mothers with less than half (40.6%) breastfeeding their babies exclusively. Married mothers were more likely to practice EBF than unmarried mothers. We, therefore, recommend policies that will improve the EBF of mothers in rural areas toward achieving the WHO target. It is also important to carry out further quantitative and qualitative studies to identify factors associated with low uptake of EBF among unmarried women and develop interventions to improve their practice.

## Supporting information

**S1 File. Minimal data set.**
(XLSX)

## Acknowledgments

We express gratitude to all the participants of this study.

## Author Contributions

**Conceptualization:** Tope Michael Ipinnimo, Olanrewaju Kassim Olasehinde, Taofeek Adedayo Sanni.

**Data curation:** Tope Michael Ipinnimo, Olanrewaju Kassim Olasehinde, Taofeek Adedayo Sanni, Ayodeji Andrew Omotoso, Rita Omobosola Alabi, Blessing Omobolanle Osho, Adewumi Rufus Fajugbagbe, Precious Aderinsola Adeyeye, Ayotomiwa Fiyinfoluwa Ajayi.

**Formal analysis:** Tope Michael Ipinnimo, Ademuyiwa Adetona, Mojoyinola Oyindamola Adeosun, Temitope Moronkeji Olanrewaju, Oluseyi Adedeji Aderinwale.

**Funding acquisition:** Tope Michael Ipinnimo, Ademuyiwa Adetona, Mojoyinola Oyindamola Adeosun, Temitope Moronkeji Olanrewaju, Oluseyi Adedeji Aderinwale.

**Investigation:** Tope Michael Ipinnimo, Olanrewaju Kassim Olasehinde, Taofeek Adedayo Sanni, Ayodeji Andrew Omotoso, Rita Omobosola Alabi, Paul Oladapo Ajayi, Kayode Rasaq Adewoye, John Olujide Ojo, Olayinka Oloruntoba, Blessing Omobolanle Osho, Adewumi Rufus Fajugbagbe, Precious Aderinsola Adeyeye, Ayotomiwa Fiyinfoluwa Ajayi.

**Methodology:** Tope Michael Ipinnimo, Olanrewaju Kassim Olasehinde, Taofeek Adedayo Sanni, Ayodeji Andrew Omotoso, Rita Omobosola Alabi, Paul Oladapo Ajayi, Kayode Rasaq Adewoye, John Olujide Ojo, Olayinka Oloruntoba, Ademuyiwa Adetona, Mojoyinola Oyindamola Adeosun, Temitope Moronkeji Olanrewaju, Oluseyi Adedeji Aderinwale, Blessing Omobolanle Osho, Adewumi Rufus Fajugbagbe, Precious Aderinsola Adeyeye, Ayotomiwa Fiyinfoluwa Ajayi.

**Project administration:** Tope Michael Ipinnimo, Olanrewaju Kassim Olasehinde, Taofeek Adedayo Sanni, Ayodeji Andrew Omotoso, Rita Omobosola Alabi, Paul Oladapo Ajayi, Kayode Rasaq Adewoye, John Olujide Ojo, Olayinka Oloruntoba, Blessing Omobolanle Osho, Adewumi Rufus Fajugbagbe, Precious Aderinsola Adeyeye, Ayotomiwa Fiyinfoluwa Ajayi.

**Resources:** Tope Michael Ipinnimo, Olanrewaju Kassim Olasehinde, Taofeek Adedayo Sanni, Ayodeji Andrew Omotoso, Rita Omobosola Alabi, Paul Oladapo Ajayi, Kayode Rasaq Adewoye, John Olujide Ojo, Olayinka Oloruntoba, Blessing Omobolanle Osho, Adewumi Rufus Fajugbagbe, Precious Aderinsola Adeyeye, Ayotomiwa Fiyinfoluwa Ajayi.

**Software:** Tope Michael Ipinnimo, Ademuyiwa Adetona, Mojoyinola Oyindamola Adeosun, Temitope Moronkeji Olanrewaju, Oluseyi Adedeji Aderinwale.

**Supervision:** Tope Michael Ipinnimo.

**Validation:** Tope Michael Ipinnimo, Olanrewaju Kassim Olasehinde, Taofeek Adedayo Sanni, Ayodeji Andrew Omotoso, Rita Omobosola Alabi, Paul Oladapo Ajayi, Kayode Rasaq Adewoye, John Olujide Ojo, Olayinka Oloruntoba.

**Visualization:** Tope Michael Ipinnimo, Olanrewaju Kassim Olasehinde, Taofeek Adedayo Sanni, Ayodeji Andrew Omotoso, Rita Omobosola Alabi, Paul Oladapo Ajayi, Kayode Rasaq Adewoye, John Olujide Ojo, Olayinka Oloruntoba, Blessing Omobolanle Osho, Adewumi Rufus Fajugbagbe, Precious Aderinsola Adeyeye, Ayotomiwa Fiyinfoluwa Ajayi.

**Writing – original draft:** Tope Michael Ipinnimo, Olanrewaju Kassim Olasehinde, Taofeek Adedayo Sanni, Ayodeji Andrew Omotoso, Rita Omobosola Alabi, Paul Oladapo Ajayi, Kayode Rasaq Adewoye, John Olujide Ojo, Olayinka Oloruntoba, Mojoyinola Oyindamola Adeosun, Blessing Omobolanle Osho, Adewumi Rufus Fajugbagbe, Precious Aderinsola Adeyeye, Ayotomiwa Fiyinfoluwa Ajayi.

**Writing – review & editing:** Tope Michael Ipinnimo, Olanrewaju Kassim Olasehinde, Taofeek Adedayo Sanni, Ayodeji Andrew Omotoso, Rita Omobosola Alabi, Paul Oladapo Ajayi, Kayode Rasaq Adewoye, John Olujide Ojo, Olayinka Oloruntoba, Ademuyiwa Adetona, Temitope Moronkeji Olanrewaju, Oluseyi Adedeji Aderinwale, Blessing Omobolanle Osho, Adewumi Rufus Fajugbagbe, Precious Aderinsola Adeyeye, Ayotomiwa Fiyinfoluwa Ajayi.

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
