## [Decision Letter · Decision Letter 0]

26 Dec 2023

PONE-D-23-40179Attitude and Predictors of Exclusive Breastfeeding Practice among Mothers Attending Under-Five Welfare Clinics in a Rural Community in Southwestern NigeriaPLOS ONE

Dear Dr. Ipinnimo,

Thank you for submitting your manuscript to PLOS ONE. After careful consideration, we feel that it has merit but does not fully meet PLOS ONE’s publication criteria as it currently stands. Therefore, we invite you to submit a revised version of the manuscript that addresses the points raised during the review process.

We look forward to receiving your revised manuscript.

Kind regards,

Kahsu Gebrekidan

Academic Editor

PLOS ONE

Journal Requirements:

3. We note that your Data Availability Statement is currently as follows: [All relevant data are within the manuscript and its Supporting Information files]

4. Please ensure that you include a title page within your main document. You should list all authors and all affiliations as per our author instructions and clearly indicate the corresponding author.

Reviewers' comments:

Reviewer's Responses to Questions

**Comments to the Author**

1. Is the manuscript technically sound, and do the data support the conclusions?

Reviewer #1: Partly

Reviewer #2: Partly

Reviewer #3: No

2. Has the statistical analysis been performed appropriately and rigorously? 

Reviewer #1: Yes

Reviewer #2: No

Reviewer #3: No

3. Have the authors made all data underlying the findings in their manuscript fully available?

Reviewer #1: No

Reviewer #2: Yes

Reviewer #3: Yes

4. Is the manuscript presented in an intelligible fashion and written in standard English?

Reviewer #1: No

Reviewer #2: No

Reviewer #3: No

5. Review Comments to the Author

Reviewer #1: Major comments

• You can include key messages in your final conclusion section

• The definition of breastmilk should be rewrite again.

• There is a reference repetition.

• This is more to the problem situation and it has to be coherent; revise the background section

• You didn't show the actual gap of the study area in this section

• The multivariate regression is don’t exist to the extent to identify the predictors

• The legend of the table description should clearly show the data

• It is good to display the multivariate data of what you have stated right here

• The discussion is not enough. You have to revise additional and similar topics in Ethiopia and other countries you have stated

In General, at this point I can give these comments.

Reviewer #2: The title has been over researched in Nigeria, so what did you add to the existing knowledge or what is your new finding? If you can justify this, what about the knowledge component? it should be included in the title or studied as a factor

The introduction section should clearly describe the public health importance of EBF and mother's attitude, it should be based on your objectives, and must show what interventions undertaken and the existing gaps

Regarding to sample size, your sample size is not adequate, and since the sampling technique is non random, it can not be used for generalization.

How do you measure attitude and practice? I think you used mean, what was your reason to use the mean? I haven't seen a clear operational definition. In addition to this using a five point Likert scale is better than 3 point Liker Scale.

It is better to show the dependent and independent variables clearly.

Regarding to ethical issue, how did you address the confidentiality issue?

Result: the statistical analysis is not performed appropriately and adequately, and lacks clarity. For example you said "The mean ± SD exclusive breastfeeding attitudinal score for the respondents was 29.94 ± 2.14 (maximum obtainable scores was 36) and the proportion of mothers that practiced exclusive breastfeeding was 40.6%." What are these numbers? showing the confidence interval is significant. The attitude score is not is not clear, you put only the mean score in table 2 there should be a table which shows the attitude components.

Marital status is the only variable associated with EBF practice, but what were the other variables in binary logistic regression which entered into multivariable logistic regression?

Discussion and conclusion should be objective based, and the recommendation should be based on your finding.

Additionally, I am not sure that the referencing style fulfils PLOS ONE's criteria.

Reviewer #3: GENERAL COMMENTS AND SUGGESTIONS

The study topic is scientifically sound and may contribute to body knowledge in maternal and child health. However, there are many gross editorial and methodological issues and problems. I have tracked and changed all the comments and suggestions with an attached document.

1. The study was written in standard academic English.

2. The methodology needs a massive correction. You could not provide a specific Analysis that fits the nature of your data.

a. You must have separate titles for data collection methods, tools, or study measurements. They are district features of the study methodology.

b. If you did a pre-test, what was the tool's reliability?

c. you need to state explicitly how you measured the exclusive breastfeeding outcome variable.

d. What is the need to use chi-square? Do you have a specific justification? You also used binary logistic regression, which measures the association between dependent and independent variables more accurately than chi-square. Table 2 also indicates you used Yate’s continuity correction and Fisher’s exact test.

e. Did you conclude the association between dependent and independent variables only based on the result of binary logic regression? This is not scientifically sound. Table 1 depicts. You used Analysis of variance and T-test. So, you did not select one appropriate analysis method for this study. If you have justification, why do you use a mix of different analyses to justify it?

3. You need to add the strengths and limitations of the study and the conclusion and recommendation sections.

6. PLOS authors have the option to publish the peer review history of their article (what does this mean?). If published, this will include your full peer review and any attached files.

Reviewer #1: No

Reviewer #2: No

Reviewer #3: No

---

## [Author Response · Author response to Decision Letter 0]

15 Jan 2024

Dear Academic Editor and Reviewers, 

Thank you for taking the time to read and review this work, we are grateful for your insightful comments and contributions. Please find the response to each point raised within the uploaded rebuttal letter.

---

## [Decision Letter · Decision Letter 1]

8 Feb 2024

PONE-D-23-40179R1Attitude and Predictors of Exclusive Breastfeeding Practice among Mothers Attending Under-Five Welfare Clinics in a Rural Community in Southwestern NigeriaPLOS ONE

Dear Dr. Ipinnimo,

Thank you for submitting your manuscript to PLOS ONE. After careful consideration, we feel that it has merit but does not fully meet PLOS ONE’s publication criteria as it currently stands. Therefore, we invite you to submit a revised version of the manuscript that addresses the points raised during the review process.

We look forward to receiving your revised manuscript.

Kind regards,

Kahsu Gebrekidan

Academic Editor

PLOS ONE

Additional Editor Comments:

The reviewer commented that the pevious comments are not fully addressed, you are expected to clearly address all comments.

Reviewers' comments:

Reviewer's Responses to Questions

**Comments to the Author**

1. If the authors have adequately addressed your comments raised in a previous round of review and you feel that this manuscript is now acceptable for publication, you may indicate that here to bypass the “Comments to the Author” section, enter your conflict of interest statement in the “Confidential to Editor” section, and submit your "Accept" recommendation.

Reviewer #1: All comments have been addressed

Reviewer #3: (No Response)

2. Is the manuscript technically sound, and do the data support the conclusions?

Reviewer #1: Partly

Reviewer #3: No

3. Has the statistical analysis been performed appropriately and rigorously? 

Reviewer #1: Yes

Reviewer #3: No

4. Have the authors made all data underlying the findings in their manuscript fully available?

Reviewer #1: Yes

Reviewer #3: (No Response)

5. Is the manuscript presented in an intelligible fashion and written in standard English?

Reviewer #1: No

Reviewer #3: No

6. Review Comments to the Author

Reviewer #1: More or less, the authors addressed the comments, and the author amended them accordingly. There are some modifications that the author didn’t consider changing. The introductory sentence should be included in the background data.

There is still a gap in justifying the actual study area.

Results and discussion

It is better to report the mean age of mothers

One of the variables in the result section is work schedule, and the majority of the respondents are farmers. How do you justify this? Similar comments were made during the discussion as well.

Reviewer #3: General comments and suggestions.

The comments and suggestions are not adequately addressed. There are areas that have needed major revision. For instance, the Methodology, result, discussion, and recommendations sections. Additionally, there are editorial issues.

Methodology: Variable page 8. You treat Attitude as the dependent variable in the result section. See Table 2. It is so confusing.

Methodology: Data management and statistical analysis. Yes, you calculated the mean and standard deviation based on the attitudinal score for each independent variable. The question is: Is attitude toward breastfeeding not a dependent variable? The second question is, do you understand when we can use T-test and ANOVA? Because to use a T-test, you need to have at least two Mean scores. For Anova, it is supposed to be three means or more. You have a single population. You collected data at a single point. You did not have a standard attitudinal score to compare with. So, your statistical analysis selection reflects a poor understanding of this method. I Attached a document for your understanding.

Result Page 10. The logical order of the paragraphs is not correct. Some information is fabricated.

Discussion: page 10 paragraph 6. EBF attitude was not stated as the dependent variable in the methodology section. So, it is difficult to discuss here. Additionally, you used a T-test to identify associated factors to EBF attitude, which is an inappropriate and unacceptable choice of analysis. Review this thoroughly. (for reference table 2.)

Discussion: page 11 paragraph 2. Where did you find this finding? Education is not in the final model. Education is not significantly associated with the practice of exclusive breastfeeding in the Chi-square test (Reference Table 3)

Discussion: page 11 paragraph 3. First, discuss descriptive findings before analytic ones. Follow the logical order.

Discussion: page 11 paragraph . A logistic regression model with a single variable cannot predict the practice of exclusive breastfeeding. This is not utterly acceptable. (reference table 4)

Conclusion and recommendation: page 12 paragraph1. These are descriptive findings from this study. However, factors associated with exclusive breastfeeding are more reliable and credible for recommending solutions. It is not conclusive to recommend unless these factors are statistically significant with exclusive breastfeeding practice. (reference Figure 1) Hence, this study's analytical recommendation should come from analytical findings.

7. PLOS authors have the option to publish the peer review history of their article (what does this mean?). If published, this will include your full peer review and any attached files.

Reviewer #1: No

Reviewer #3: No

---

## [Author Response · Author response to Decision Letter 1]

12 Feb 2024

Dear Academic Editor and Reviewers,

Thank you once again for your comments and contributions. Explanations have been given and reference have been provided for the statistical method used in analysis. Please find the response to each point raised within the uploaded rebuttal letter.

Regards

---

## [Decision Letter · Decision Letter 2]

19 Feb 2024

Attitude and Predictors of Exclusive Breastfeeding Practice among Mothers Attending Under-Five Welfare Clinics in a Rural Community in Southwestern Nigeria

PONE-D-23-40179R2

Dear Mr Tope MIchael,

We’re pleased to inform you that your manuscript has been judged scientifically suitable for publication and will be formally accepted for publication once it meets all outstanding technical requirements.

Kind regards,

Kahsu Gebrekidan

Academic Editor

PLOS ONE

Additional Editor Comments (optional):

Reviewers' comments:

Reviewer's Responses to Questions

**Comments to the Author**

1. If the authors have adequately addressed your comments raised in a previous round of review and you feel that this manuscript is now acceptable for publication, you may indicate that here to bypass the “Comments to the Author” section, enter your conflict of interest statement in the “Confidential to Editor” section, and submit your "Accept" recommendation.

Reviewer #3: All comments have been addressed

2. Is the manuscript technically sound, and do the data support the conclusions?

Reviewer #3: Yes

3. Has the statistical analysis been performed appropriately and rigorously? 

Reviewer #3: Yes

4. Have the authors made all data underlying the findings in their manuscript fully available?

Reviewer #3: Yes

5. Is the manuscript presented in an intelligible fashion and written in standard English?

Reviewer #3: Yes

6. Review Comments to the Author

Reviewer #3: Authors addressed all comments and suggestion. This study will contribute a significant view toward maternal and child health body of knowledge specially in the area limited research has been conducted.

7. PLOS authors have the option to publish the peer review history of their article (what does this mean?). If published, this will include your full peer review and any attached files.

Reviewer #3: No

---

## [Editor Report · Acceptance letter]

19 Mar 2024

PONE-D-23-40179R2 

PLOS ONE

Dear Dr. Ipinnimo, 

I'm pleased to inform you that your manuscript has been deemed suitable for publication in PLOS ONE. Congratulations! Your manuscript is now being handed over to our production team.

Kind regards, 

on behalf of

Dr. Kahsu Gebrekidan 

Academic Editor

PLOS ONE